# The Role of Anti-Inflammatory Adipokines in Cardiometabolic Disorders: Moving beyond Adiponectin

**DOI:** 10.3390/ijms222413529

**Published:** 2021-12-16

**Authors:** Han Na Jung, Chang Hee Jung

**Affiliations:** 1Asan Medical Center, Department of Internal Medicine, University of Ulsan College of Medicine, Seoul 05505, Korea; wgilrw@naver.com; 2Asan Diabetes Center, Asan Medical Center, Seoul 05505, Korea

**Keywords:** adipokines, adiponectin, vaspin, CTRP9, SFRP5, omentin-1

## Abstract

The global burden of obesity has multiplied owing to its rapidly growing prevalence and obesity-related morbidity and mortality. In addition to the classic role of depositing extra energy, adipose tissue actively interferes with the metabolic balance by means of secreting bioactive compounds called adipokines. While most adipokines give rise to inflammatory conditions, the others with anti-inflammatory properties have been the novel focus of attention for the amelioration of cardiometabolic complications. This review compiles the current evidence on the roles of anti-inflammatory adipokines, namely, adiponectin, vaspin, the C1q/TNF-related protein (CTRP) family, secreted frizzled-related protein 5 (SFRP5), and omentin-1 on cardiometabolic health. Further investigations on the mechanism of action and prospective human trials may pave the way to their clinical application as innovative biomarkers and therapeutic targets for cardiovascular and metabolic disorders.

## 1. Introduction

Obesity has been a growing threat worldwide owing to its association with various cardiometabolic disorders, including insulin resistance, type 2 diabetes mellitus (T2DM), fatty liver disease, and cardiovascular disease [1]. Accumulated evidence suggests that obesity-induced inflammation plays a key role in the pathogenesis of these disorders [1,2,3]. Obesity initiates chronic low-level inflammation, possibly by means of increased expression of pro-inflammatory genes and oxidative stress [2].

Contrary to the traditional concept of adipose tissue as an inert storage site for redundant energy, it is now highlighted as an endocrine organ that is involved in metabolic homeostasis. Since the characterization of leptin in 1994 [4], a number of adipocyte-derived compounds have been identified. These substances known as adipokines modulate appetite control, energy expenditure, insulin secretion, cardiovascular function, inflammation, immunity, adipogenesis, reproduction, and bone metabolism [5]. The inflammation and dysfunction of adipose tissue induce an alteration in adipokine secretion toward a pro-inflammatory, diabetogenic, and atherogenic pattern. Adipokines act not only locally by mediating the crosstalk between adipocytes, endothelial cells, and macrophages but also at the systemic level [2].

Most adipokines, including leptin, tumor necrosis factor-alpha (TNF-α), interleukin 6 (IL-6), resistin, or retinol-binding protein 4 (RBP4), are augmented in the obese state and promote inflammatory reactions, leading to obesity-associated comorbidities [6]. Contrarily, a smaller number of adipokines are produced by metabolically healthy adipose tissue to decrease inflammation and have protective effects on metabolic dysfunction [1]. This review focuses on the beneficial role of several anti-inflammatory adipokines of our interest in cardiometabolic disorders.

## 2. Adiponectin

### 2.1. Biological Actions

Adiponectin is one of the most extensively investigated adipokines since its first description in 1995 [7]. It is primarily produced by white adipose tissue, although additional sources, such as skeletal muscles or cardiomyocytes, have been known [8,9]. Circulating adiponectin exists in three different oligomers: low-molecular-weight trimer, middle-molecular-weight hexamer, and high-molecular-weight (HMW) multimer. HMW adiponectin is the principal bioactive form in metabolic tissues [10]. Activation of the two recognized receptors for adiponectin, AdipoR1 and R2, is linked to the stimulation of AMP-activated protein kinase (AMPK), p38 mitogen-activated protein kinase, and peroxisome proliferator-activated receptor alpha (PPARα) [11].

Adiponectin suppresses M1, which induces inflammation and insulin resistance while activating M2, which intensifies the anti-inflammatory response and oxidative metabolism [12,13,14]. Anti-inflammatory cytokines such as IL-10 are stimulated, and pro-inflammatory factors including TNF-α, interferon-gamma (IFN-γ), and vascular cell adhesion molecule-1 (VCAM-1) are inhibited by adiponectin [15,16]. Meanwhile, adiponectin inhibits hepatic glucose production, as well as promotes glucose utilization and fatty acid oxidation of the skeletal muscles via the stimulation of the AMPK and PPARα pathway [17,18,19,20] (Figure 1). Cytokine- or ceramide-induced β cell apoptosis is blocked by adiponectin, further contributing to insulin sensitivity [21,22]. Moreover, augmented nitric oxide (NO) synthesis by AMPK following the adiponectin-mediated phosphorylation of endothelial nitric oxide synthase (eNOS) attenuates vasoconstriction and progression of atherosclerosis [23,24]. Adiponectin also interferes with vascular smooth muscle cell (VSMC) hypertrophy [25]. Collectively, adiponectin has anti-inflammatory, insulin-sensitizing, and anti-atherogenic effects.

Several anti-diabetic medications are also recognized to intensify adiponectin expression (Figure 1). For instance, pioglitazone not only upregulates the adiponectin levels in adipose tissue but also affects the multimerization of adiponectin into the potent, HMW form [26]. Likewise, glucagon-like peptide-1 (GLP-1) analogs directly enhance the secretion of adiponectin via protein kinase A (PKA) signaling in vitro [27]. Lastly, T2DM mice administered with sodium-glucose cotransporter-2 (SGLT2) inhibitors exhibit elevation of plasma adiponectin levels [28]. Although further studies are needed, adiponectin may be the mediator of the cardiovascular benefits of pioglitazone, GLP-1 analogs, and SGLT2 inhibitors, which have been established by randomized controlled trials [29,30,31].

Recently, adiponectin is suggested to be a downstream mediator of fibroblast growth factor (FGF) 21, which is another potent modulator of glucose and lipid metabolism produced in adipose tissue [32]. The expression of adiponectin is stimulated by the acute injection of FGF21 through activating peroxisome proliferator-activated receptor gamma (PPARγ) [33]. In addition, the resistance to FGF21 inhibits adiponectin secretion [34]. Laboratory studies have advocated that adiponectin mediates the metabolic effects of FGF21, although the direct mechanism needs to be specified.

### 2.2. Human Relevance

Plasma adiponectin levels are reduced in patients with obesity [35], T2DM [36,37], atherosclerosis [38], or hypertension [39]. Clinical studies have shown that the VLDL and HDL levels are inversely and positively correlated with the adiponectin levels, respectively [40,41]. A recent cohort study proposed that low adiponectin levels may predict the diminished capability of cholesterol efflux regardless of body mass index (BMI) [42]. Along with an inverse relationship to total fat mass [35], adiponectin secretion is also regulated by the quality of adipose tissue [43]. Metabolically healthy but obese individuals tend to have higher levels of adiponectin compared with their unhealthy counterparts with a similar amount of adipose tissue [44]. In addition, the disorganized formation of adiponectin isoforms may be associated with cardiometabolic disorders. Patients with coronary artery disease (CAD) show a lower proportion of HMW multimer in contrast to a higher trimeric form. Likewise, only HMW adiponectin is increased following weight loss in obese patients [45]. The augmentation of adiponectin action through increasing its level or enhancing the signaling pathways may be a therapeutic approach for cardiometabolic diseases. Human levels of circulating adiponectin are positively affected by weight loss with a low-calorie diet [46]; medications, such as sibutramine or phentermine/topiramate [47,48]; and bariatric surgery [49].

Recently, some studies emphasized the contradictory results indicating that high adiponectin levels are associated with unfavorable cardiovascular and other metabolic outcomes [50,51,52]. This contradiction, referred to as the adiponectin paradox, might be attributed to a compensatory response in cardiovascular disorders, instead of precipitating them. Reduced adiponectin clearance is another possible explanation considering that hyperadiponectinemia is frequently observed in patients who have both cardiovascular diseases and hepatic or renal damage [43]. Finally, adiponectin isoforms rather than concentration may have a detrimental role in cardiovascular homeostasis. Even though the adiponectin paradox makes it cumbersome to apply adiponectin as a biomarker, it is still a promising therapeutic target for cardiometabolic disorders.

## 3. Vaspin

### 3.1. Biological Actions

Visceral adipose tissue-derived serpin protease inhibitor (vaspin) belongs to serpin family A member 12 (serpin A12) in accordance with the serpin nomenclature [53]. It was first isolated from the Otsuka Long-Evans Tokushima fatty (OLETF) rat, a T2DM animal model with obesity [54]. Vaspin is expressed in not only visceral and subcutaneous adipose tissues but also other organs, including the pancreas, liver, stomach, and skin [55]. Vaspin binds to a 78 kDa glucose-regulated protein and initiates multiple kinase signaling pathways such as AMPK and Akt [56]. Two protease targets of vaspin have been discovered so far, that is, kallikrein 7 and kallikrein 14, which are inhibited by vaspin through the classical serpin mechanism [57,58].

Vaspin has beneficial effects on glucose metabolism, vascular health, appetite control, and lipid profile (Figure 2). Vaspin expression in adipose tissue and plasma vaspin concentration increases in the highest point of insulin resistance and obesity but decreases with the aggravation of T2DM and body weight loss in the OLETF rat [54]. This conflicting tendency of vaspin contrary to adiponectin may imply the compensatory role of vaspin in metabolic dysfunction. Obviously, in vivo, the infusion of recombinant vaspin or transgenic vaspin overexpression enhances glucose tolerance and insulin sensitivity [54,59,60]. Vaspin promotes pancreatic islet cell secretion, protects β cells from nuclear factor-kappa B (NF-κB)-mediated inflammatory damage, and reduces glucose production in the liver [61,62].

Counteracting vascular inflammation is one of the major capacities of vaspin for preserving endothelial function. Vaspin attenuates NF-κB activity in response to the reduced secretion of inflammatory cytokines in the AMPK-dependent pathway [63,64,65]. The macrophage phenotype is shifted by vaspin toward anti-inflammatory M2 rather than M1 with the downregulation of NF-κB and upregulation of PPARγ [59]. Moreover, vaspin dampens the generation of reactive oxygen species (ROS) and oxidative stress-induced apoptosis of mesenchymal stem cells (MSCs) [66,67,68,69]. The progression of aortic atherosclerosis is mitigated by vaspin through impeding intimal proliferation and plaque instability [59,70]. We also demonstrated that vaspin was involved in the aortic vasorelaxation following the augmentation of eNOS activity via favorable effects on the signal transducer and activator of transcription 3 and asymmetric dimethylarginine system [71]. Furthermore, vaspin modulates feeding regulation and lipid metabolism. Both the peripheral and central application of vaspin reduce food intake [72], partly due to the downregulated expression of the hypothalamic orexigenic neuropeptides [73]. High-dose vaspin infusion decreases the free fatty acid and triglyceride levels as well as promotes cholesterol efflux via the upregulation of ATP-binding cassette transporter A1 (ABCA1) in macrophages [59,74]. These biological actions of vaspin support its role as an anti-atherogenic adipokine.

### 3.2. Human Relevance

Compared with non-obese controls, serum levels of vaspin are higher in obese subjects with even normal body weight [75]. A meta-analysis verified significantly higher vaspin concentrations in subjects with T2DM compared with non-diabetic subjects [76]. Increased vaspin concentrations in T2DM patients are also associated with the elevated risk of diabetic complications, such as CAD or diabetic retinopathy [77,78]. The vaspin levels decrease as insulin sensitivity improves through exercise, anti-diabetic medication such as metformin and rosiglitazone, and bariatric surgery [79,80,81,82]. Meanwhile, variants of the vaspin rs2236242 gene have been found to be correlated with the development of T2DM independently of obesity [83]. Contrarily, treatment with atorvastatin or rosuvastatin has been shown to increase the vaspin levels in patients with non-significant carotid atherosclerosis and acute coronary syndrome, respectively [84,85]. Low plasma vaspin levels are significantly associated with a higher risk of preclinical carotid atherosclerosis [84], acute coronary syndrome, coronary in-stent restenosis after percutaneous coronary intervention (PCI) [86], and poorer prognosis after myocardial infarction (MI) [87]. The contrasting results of the association between the vaspin levels and cardiometabolic health might have originated from the different study designs.

Vaspin has been investigated as a diagnostic tool for cardiometabolic disorders. A retrospective observational study on individuals with chest pain implied the availability of vaspin as a predictive biomarker of major adverse cardiovascular events (MACE) [88]. The vaspin expression levels significantly indicated the risk of ischemic stroke in a prospective study of T2DM patients [89]. In women with polycystic ovary syndrome (PCOS), serum vaspin concentrations can make a distinction between individuals with a higher diabetogenic risk [90]. The vaspin levels also predict metabolic syndrome as a single entity, as well as some of its components [91,92]. Taken together, beginning with the identification from the diabetic animals, vaspin is expected to be applicable in cardiometabolic diseases.

## 4. C1q/TNF-Related Protein (CTRP) Family

The CTRP family is a conserved group of adiponectin paralogs, containing a structural similarity to adiponectin. Along with the epicardial adipose tissue, which is the main secretory organ, various sources, such as the heart, liver, kidney, and muscles, have been recognized [93]. Each member of the 15 identified CTRP isoforms has a distinct function. To sum up the latest research, CTRP1, CTRP3, CTRP5, CTRP6, CTRP9, CTRP12, CTRP13, and CTRP15 engage in the development of cardiometabolic diseases, among which CTRP1 and CTRP5 promote the pro-inflammatory response, whereas the others favor the opposite [94] (Figure 3). 

### 4.1. CTRP9

#### 4.1.1. Biological Actions 

Of all its family members, CTRP9 has attracted the most attention following its discovery in 2009 [95]. CTRP9 shares the greatest homology with adiponectin involving 51% amino acid overlap [95]. In contrast to adiponectin that maintains the full length in the plasma, the major circulatory and biologically active structure of CTRP9 is the globular form produced by the proteolytic cleavage of trimeric complexes [96]. The heart is the third richest organ for CTRP9 distribution, and cardiac function is significantly influenced by CTRP9, making CTRP9 not only an adipokine but also a cardiokine [97]. Currently, two receptors have been introduced: AdipoR1 and N-cadherin, which is a cell surface marker of MSCs [98,99]. 

The administration of CTRP9 exerts anti-inflammatory effects by downregulating TNF-α with the downstream mediators through the activation of the AdipoR1/AMPK/NF-kB signaling cascade [100,101]. In addition, it interferes with macrophage polarization, stimulating the M1 to M2 transition [102]. The targeted deletion of murine CTRP9 leads to obesity, insulin resistance, and hepatic steatosis through both the central and peripheral mechanisms with the overexpression of orexigenic peptides in the hypothalamus, blunting hepatic insulin signaling, and upregulating the lipogenic genes [103]. Recombinant globular CTRP9 elevates the HDL levels while reducing the LDL and triglyceride levels, via the enhancement of cholesterol efflux by accelerating AdipoR1/AMPK signaling and inhibition of lipoprotein uptake by suppressing lectin-like oxidized low-density lipoprotein receptor 1 [100,104,105,106]. In vivo and in vitro studies have shown that the inhibitory effect of CTRP9 on VSMC proliferation, neointimal hyperplasia, and platelet activation prevents the progression of atherosclerosis [107,108,109]. The trans-differentiation of VSMCs into macrophage-like cells aggravates the inflammatory and proliferative response as well as debilitates the vascular contractile function, which is antagonized by CTRP9 [100]. Furthermore, CTRP9 serves as a defender of endothelial function by facilitating vascular relaxation and ischemia-induced revascularization through the AMPK/Akt/eNOS pathway [110,111,112]. It downregulates ROS production and apoptosis of endothelial cells [113,114,115], along with promoting the phagocytic removal of apoptotic cells to deter subsequent necrosis and inflammation [116]. Likewise, we have shown that palmitic acid-dependent endothelial senescence, which is a significant risk factor of vascular atherosclerosis, was abrogated by CTRP9-mediated restoration of AdipoR1-/AMPK-mediated autophagy [117]. In addition, the anti-fibrotic action of CTRP9 alleviates the post-MI pathologic cardiac remodeling and ventricular dysfunction [118]. Altogether, these actions of CTRP9 contribute to anti-inflammation and anti-atherogenesis.

#### 4.1.2. Human Relevance

Two retrospective clinical studies on the association of the CAD and CTRP9 levels presented conflicting results [119,120]. Nonetheless, numerous human data favor the protective effects of CTRP9 on the cardiovascular system. Serum levels of CTRP9 along with CTRP3 are reduced in proportion to the severity of heart failure with reduced ejection fraction [121]. In patients who underwent kidney transplantation, plasma CTRP9 concentration is an improving factor of the aortic calcification [122]. In addition, elevated CTRP9 levels are correlated with less restenosis following the implantation of cerebrovascular stent [123].

Similar to coronary atherosclerosis, antecedent reports on plasma CTRP9 concentrations in patients with metabolic syndrome showed inconsistent trends [124,125]. Obese patients are associated with higher CTRP9 levels, which are downregulated following bariatric surgery [126]. Conversely, serum CTRP9 levels are significantly lower in patients with gestational diabetes mellitus (GDM) [127]. It is cautiously supposed that the CTRP9 levels fluctuate to compensate in the early stage of metabolic dysfunction, taking into account that the CTRP9 expression is increased at 8 weeks of age but decreased 4 weeks later in leptin-deficient obese mice [95]. Several clinical studies of diabetic patients suggested plasma CTRP9 levels as reliable biomarkers for cardiac autonomic neuropathy, carotid intima-media thickness (IMT), and arterial stiffness represented by brachial-ankle pulse wave velocity (baPWV) [128,129,130]. Likewise, CTRP9 levels are independent risk factors for the aggravation of GDM [131].

### 4.2. Other Members of the CTRP Family with Anti-Inflammatory Effects

#### 4.2.1. Biological Actions

In conjunction with the most renowned CTRP9, other anti-inflammatory isoforms, such as CTRP3, CTRP12, and CTRP13, exert insulin-sensitizing, anti-atherosclerotic, and cardioprotective properties. CTRP3 remarkably decreases the inflammatory cytokines, suppresses cellular apoptosis, and equilibrates the opposing effects of endothelin-1 and NO by prompting the phosphatidylinositol 3-kinase (PI3K), Akt, and eNOS expressions [132]. CTRP12, also referred to as adipolin, is inversely correlated with IL-6 and TNF-α, and encourages the M2 phenotype of macrophage polarization [133,134]. CTRP3 inhibits hepatic glucose output [135], and both CTRP12 and CTRP13 repress gluconeogenesis, intensify the glucose uptake of various organs, and relieve hepatic insulin resistance [136,137]. Additionally, CTRP12 disturbs the neointimal thickening by alleviating vascular cell proliferation and lipid accumulation following the promotion of ABCA1-mediated cholesterol efflux [134,138]. CTRP13-infused mice are protected from atherosclerotic plaque formation with accelerated autophagy, downregulated lipid uptake, and inhibited foam cell migration [139].

Post-MI cardiac impairment is mitigated with the overexpression of CTRP3 by preserving the survival, migration, and antioxidant capacity of MSCs [140]. The anti-fibrotic, anti-apoptotic, and angiogenic functions of CTRP3 further ameliorate adverse cardiac remodeling after MI [141,142]. Conversely, CTRP3 is activated in dysfunctional human hearts, and it rather develops cardiac hypertrophy in response to overpressure [143]. This discrepancy may have derived from different trial models, which needs to be further elucidated. For CTRP12, the beneficial effects on myocardial survival and ventricular contractile function have been confirmed recently [144].

#### 4.2.2. Human Relevance

A longitudinal study demonstrated that the circulating CTRP3 levels are negatively related to obesity, non-alcoholic fatty liver disease, and the indicators of pancreatic β-cell function [145]. Reflecting the compensative role, serum CTRP3 levels are increased in the pre-diabetic state but decreased in T2DM patients [146,147]. Metformin treatment upregulates the concentrations of CTRP3 and CTRP12 in humans [148,149]. Likewise, the CTRP13 levels present an inverse relationship with the adverse metabolic factors including BMI, insulin, lipidemia, and carotid IMT [150].

The CTRP3 levels are reduced in patients with CAD, acute aortic dissection, and heart failure [151,152,153]. Similarly, the CTRP12 levels are negatively correlated with the severity of CAD [154]. In the meantime, complement 1q (C1q), a complement subcomponent that provokes the classical immune pathway, can bind to adiponectin [155]. Plasma C1q-adiponectin/total adiponectin ratio is positively associated with atherosclerosis measured by vascular ultrasonography and can serve as an indicator of CAD [156,157,158,159].

To date, the anti-inflammatory members of the CTRP family show a generally compatible contribution to cardiovascular and metabolic homeostasis, despite the inconsistent findings in some studies.

## 5. Secreted Frizzled-Related Protein 5 (SFRP5)

### 5.1. Biological Actions

Wingless-type family member 5a (WNT5A), a glycopeptide secreted by adipose tissue macrophages, belongs to a non-canonical member of the Wnt family [160]. WNT5A has been implicated in cellular inflammation, proliferation, and migration to notably develop atherosclerosis and insulin resistance [6,161]. In persistent low-grade inflammation, the actions of WNT5A are offset by SFRP5 which is produced by healthy adipose tissues [162] (Figure 4). SFRP5 is one of the five identified SFRP families, which is the largest group of Wnt inhibitors. According to Ouchi et al., SFRP5 is substantially expressed in white adipose tissue in addition to other insulin targets, including the liver, muscle, and beta cells [163]. Although both the canonical and non-canonical Wnt pathways are impeded by SFRP5, the blockage of WNT5A signaling has been established to play a dominant role in the modulation of metabolic homeostasis [164,165].

The anti-inflammatory function of SFRP5 is exerted by the neutralization of c-Jun N-terminal kinases (JNK) in the downstream of the Wnt signaling pathway, to repress the production of inflammatory cytokines [162,166] (Figure 4). SFRP5 facilitates adipogenic differentiation and adipocyte expansion [167,168]. Despite the augmented expression of SFRP5 in adipose tissues, it seems to decline following the aggravation of metabolic dysfunction, considering the transient upregulation of SFRP5 with milder inflammation in adipocytes and ensuing drop after the development of obesity. In the same way, SFRP5 knockout mice acquire severe insulin resistance and hepatic steatosis with a greater macrophage accumulation in their adipocytes [162]. SFRP5 suppresses beta cell proliferation but enhances glucose-dependent insulin production, indicating the protective role on beta cell function [169]. Moreover, the central infusion of SFRP5 in vivo ameliorates glucose metabolism through brain-hepatic circuits, with less food intake, more energy expenditure, and potentiated hypothalamic neurons engaged in insulin activity, while inhibiting hepatic gluconeogenesis and lipogenesis via the hepatic vagus nerve [170].

The upregulation of SFRP5 significantly attenuates apoptosis of human endothelial cells, whereas the overexpression of WNT5A or knockdown of SFRP5 exhibits the opposite outcomes [171,172]. We have previously substantiated that SFRP5 dose dependently reverses impaired vasorelaxation by increasing the NO production via blockage of the WNT5A/JNK pathway [173]. SFRP5 also restricts VSMC proliferation and migration as well as decreases ROS generation in the murine aorta [174]. Meanwhile, SFRP5 exerts the restoring function in ischemic heart disease, as evidenced by SFRP5-deficient mice presenting more extensive inflammation, apoptosis, infarct size, and cardiac dysfunction consequent to ischemia/reperfusion injury [175]. SFRP5 also interferes with the survival, proliferation, and migration of cardiac fibroblasts [176].

### 5.2. Human Relevance

Obesity predisposes to decreased SFRP5 and increased WNT5A expression, which has been confirmed in human cohorts with atherosclerosis [177]. Cross-sectional studies have demonstrated that higher SFRP5 levels are negatively associated with odds of prediabetes and T2DM, along with multiple risk factors, such as systolic blood pressure, BMI, and C-reactive protein levels [178,179]. Likewise, lifestyle intervention with significant weight loss and anti-diabetic medications, including metformin and liraglutide, increase the SFRP5 levels in humans [180,181,182]. Numerous clinical trials showed the association between higher SFRP5 levels and a favorable lipid profile [178,179,183,184]. Contrarily, another study suggested that elevated SFRP5 concentrations are rather correlated with a higher risk of T2DM [185]. Further investigation is needed to determine whether the SFRP5 levels vary according to the progression of metabolic dysfunction.

Corresponding to in vitro and in vivo studies, clinical trials have indicated the salutary effects of SFRP5 on vascular health. In the adipocytes of patients with the peripheral arterial disease (PAD), the plasma concentrations of WNT5A are significantly increased but those of SFRP5 are significantly decreased [186]. Similarly, the serum SFRP5 levels are lower in chronic kidney disease subjects with vascular calcification compared with those without calcification [187]. On the other hand, we reported the positive correlation between SFRP5 concentrations and baPWV based on the multivariate linear regression analysis of T2DM patients. This discrepancy in the SFRP5 levels may be interpreted as a compensatory rise in incipient atherosclerosis, taking into account the restoration of endothelial NO synthesis by SFRP5 [173].

Human data concerning the role of SFRP5 on CAD have been discordant. CAD patients have higher plasma WNT5A levels and lower SFRP5 levels compared with those without CAD, independent of the conventional risk factors [177]. An experiment using the mRNA levels of SFRP5 and WNT5A from epicardial adipose tissue biopsies of CAD patients reconfirmed the results [188]. Additionally, the WNT5A concentration is independently correlated with the occurrence and advancement of coronary calcification [177]. In the acute phase of ST-segment elevation MI, high SFRP5 concentrations are significantly related to early myocardial recovery after primary PCI [189]. Furthermore, a prospective cohort study proposed SFRP5 as a novel prognostic marker for heart failure on the basis of enhanced discrimination by conjoining SFRP5 to the traditional risk factors [190]. On the contrary, elevated SFRP5 levels predicted the incidence of MACE in a 4-year prospective study [191]. In spite of the conflicting outcomes regarding the relationship of the SFRP5 levels with insulin resistance and CAD, the majority of up-to-date research supports the beneficial role of SFRP5 in cardiometabolic health. Future research on the temporal variation of SFRP concentration during the disease course as well as the mechanism of action is required.

## 6. Omentin-1

### 6.1. Biological Actions

Omentin is a novel adipokine comprising 313 amino acids [192]. It is preferentially produced by visceral adipose tissue rather than subcutaneous adipose tissue. Omentin-1 and omentin-2 are two identified isoforms showing analogous gene expression, among which omentin-1 is a dominant form in the human circulation [193]. Cumulative experimental studies have demonstrated the direct intervention of omentin-1 in anti-inflammation, insulin sensitization, and cardiovascular health mainly through the cascade of AMPK/Akt/NF-kB/extracellular signal-regulated kinase (ERK), JNK, and p38 [194]. The antagonistic effects of omentin-1 on inflammation and atherosclerosis have been confirmed by various in vivo and in vitro studies. Transgenic mice with the human omentin gene show decreased expression of pro-inflammatory cytokines via the stimulation of Akt [195]. Omentin-1 also suppresses macrophage aggregation, apoptosis, foam cell formation, and M1 phenotype [195,196,197]. Previously built atherosclerotic lesions are diminished by omentin-1 through reducing lipid content and necrotic core [195,196]. Neointimal hyperplasia and VSMC proliferation are attenuated by omentin via the AMPK/ERK pathway [198]. Likewise, omentin-1 overexpression mitigates arterial calcification through PI3K/Akt-dependent mechanisms [199].

Cardiovascular dysfunction is also ameliorated by omentin-1. Bioavailability of NO is improved by omentin-mediated phosphorylation of eNOS to promote vasorelaxation and revascularization in contracted artery [200,201,202]. Omentin-derived activation of the canonical pathway of AMPK/Akt/ERK attenuates myocyte apoptosis, hypertrophy, and fibrosis following myocardial ischemia [203,204]. Similarly, overexpression of omentin-1 enhances parameters of cardiac function and cardiac mitochondrial activities possibly through upregulating glycogen synthase kinase-3 beta pathway in the rat model [205].

### 6.2. Human Relevance

Investigations on the correlation of human plasma levels of omentin-1 and cardiometabolic disorders display remarkably consistent results. Cross-sectional studies determined significantly lower omentin-1 levels in obese individuals compared with lean controls [193,206]. Even after adjusting for BMI, both serum and adipocyte levels of omentin-1 are inversely associated with the presence of metabolic syndrome [207]. A negative correlation of the serum omentin-1 levels with T2DM and GDM has also been reported [206,208]. Moreover, plasma omentin-1 is further reduced in T2DM patients with diabetic complications including retinopathy, peripheral neuropathy, nephropathy, and PAD than in those without complications [209,210]. In addition to decreased omentin-1 levels, patients with diabetic nephropathy exhibit significantly increased IL-6 which plays a critical role in immune response [211]. Serum IL-6 levels are independent factors determining omentin-1 levels, indicating the crosstalk between the adipose tissue and immune system [212].

Decreased plasma omentin-1 levels are not only significantly associated with CAD but also reflective of the disease severity, with lower values in acute coronary syndrome than in stable angina pectoris [213,214]. Biscetti et al. performed a prospective cohort study with T2DM subjects who have PAD and chronic limb-threatening ischemia to reveal that declined omentin-1 levels at baseline are related to higher risks of MACE and major adverse limb events [215]. On the other hand, higher plasma omentin levels are associated with enhanced cardiac function in patients with MI and heart failure [203,216]. Omentin-1 is also thought to influence blood pressure and lipid profile regulation. Individuals with stage 1 and 2 hypertension have significantly lower omentin-1 in circulation compared with their normotensive counterparts [217]. HDL and VLDL concentrations are correlated with omentin-1 levels positively and negatively, respectively [193,207].

The possibility of using circulating omentin-1 levels as biomarkers for cardiovascular diseases has been raised. Reduced baseline levels of omentin-1 predict worse functional prognosis, carotid plaque instability, and 1-year mortality in patients with acute ischemic stroke [218,219,220,221], as well as poorer clinical outcomes for acute intracerebral hemorrhage [222]. Lower omentin-1 levels are also prognostic markers of frequent MACE, severe angiographic CAD, and incomplete coronary collateral circulation [215,223,224]. In addition, plasma omentin-1 levels are independent determinants of several vascular complications of diabetic patients including carotid plaque, arterial stiffness measured by baPWV, PAD, and retinopathy [210,225,226,227].

Upregulation of circulating omentin-1 levels may be a novel therapeutic approach to delay the progression of cardiometabolic diseases. A recent prospective cohort study of obese or overweight children discovered that both short-term and long-term lifestyle interventions correlated positively with omentin-1 levels [228]. Similarly, a high-intensity exercise program of 12-week duration and weight loss after biliopancreatic diversion surgery augments plasma omentin-1 levels [229,230]. The widely prescribed medications such as metformin and atorvastatin are also expected to increase serum levels of omentin-1 [84,231]. In conclusion, coherent results of animal and human data on the favoring effect of omentin-1 on the cardiometabolic system may allow omentin-1 to be developed as a predictive marker or therapeutic medium.

## 7. Conclusions

Obesity, beyond its definition indicating the static state of excessive fat accumulation, accompanies chronic low-grade inflammation and is associated with a broad spectrum of cardiometabolic diseases, including T2DM, dyslipidemia, CAD, and fatty liver disease. Likewise, the pleiotropic effects of adipokines on metabolic homeostasis have also been investigated. Together with the autocrine and paracrine actions on local adipose tissue and macrophages, adipokines reach the systemic vasculature to actively interact with the central nervous system, pancreatic islets, or liver. This review specifically summarizes the cardiometabolic function of anti-inflammatory adipokines, covering from prototypical adiponectin to the more recently discovered omentin-1. The present review is distinguished by its explicit partition of experimental and clinical data as well as the introduction of diverse, up-to-date research on humans, especially regarding the applicability as biomarkers and therapeutic targets. These advantages allow for a better understanding of current knowledge and constraints, along with an outline of future study.

A number of retrospective clinical studies have implied the predictive value of adipokines for the risk of cardiovascular disorders and the capability as biomarkers replacing more complicated diagnostic tests. Contrary to the consistent tendency presented in the animal data, human trials showed incompatible results of the associations between the serum adipokine levels and diseases (Table 1). More mechanistic explanations and prospective human trials are needed to overcome the limitations of animal studies and contradictory outcomes. The exploration of the dynamic expression of adipokines, rather than the single serum concentration, will confirm their role as prognostic markers for cardiovascular diseases. Additionally, further clinical investigation concerning the relationship of demographic factors such as age, sex, and ethnicity with the cardiometabolic effects of adipokines may be valuable. Furthermore, expanding from the gathered animal data indicating that the upregulation or exogenous administration of each substance exhibits therapeutic effects on vascular and metabolic function, the manipulation of the axis connecting the anti-inflammatory adipokines and target organs may be the blueprint for the future approaches in managing cardiometabolic disorders. Reasonable strategies for applying adipokines as biotargets are manufacturing the analogs which are stable in the circulation to sustain enough therapeutic concentration, in addition to influencing directly the action target and downstream signaling.

## Figures and Tables

**Figure 1 ijms-22-13529-f001:**
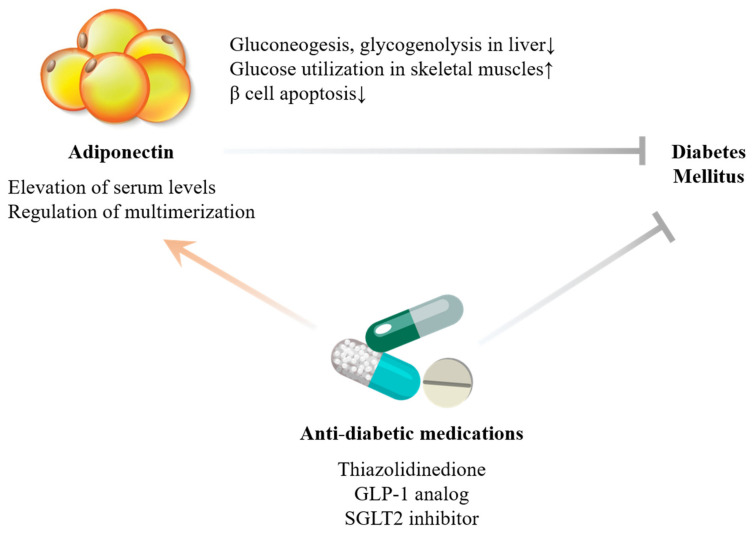
Anti-diabetic potential of adiponectin. Adiponectin enhances insulin sensitivity through multiorgan involvement to alleviate the progression of diabetes mellitus. Cumulative evidence has supported the favorable actions of anti-diabetic medications on adiponectin. The up arrows denote the increase, while the down arrows denote the opposite.

**Figure 2 ijms-22-13529-f002:**
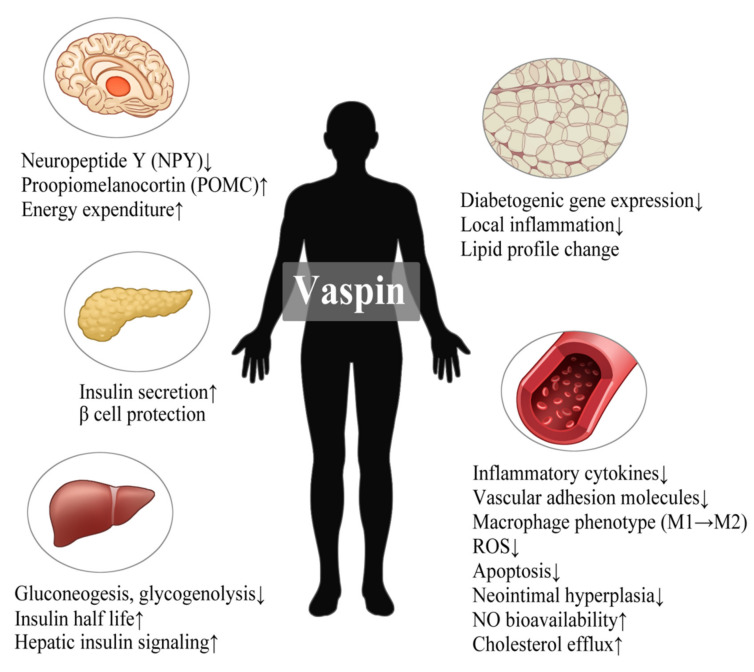
Beneficial effects of vaspin on metabolic health. Vaspin affects a broad range of organs simultaneously to enhance metabolic health. Aside from its anti-inflammatory function, vaspin intervenes in energy regulation, glycemic control, lipid profile, and vascular function. The up arrows denote the increase, while the down arrows denote the opposite.

**Figure 3 ijms-22-13529-f003:**
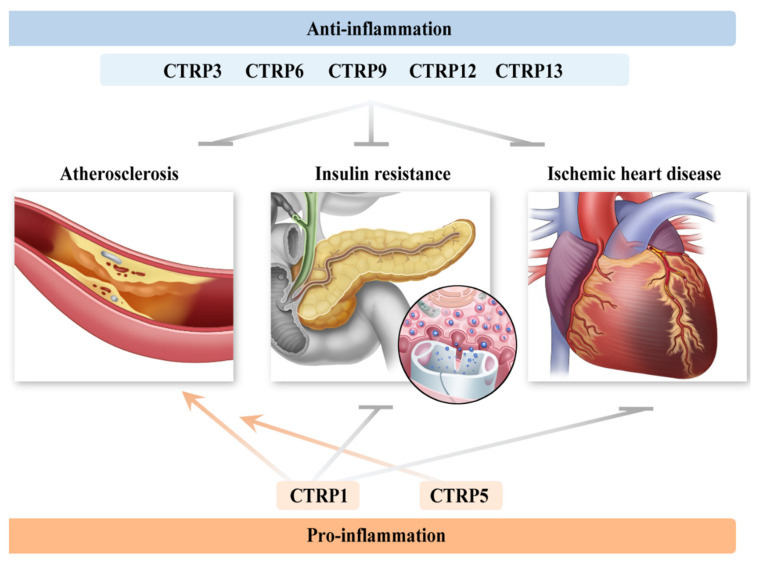
Major C1q/TNF-related protein (CTRP) isoforms participating in cardiometabolic homeostasis. The respective members of the CTRP family exert different actions on each organ, among which CTRP3, CTRP6, CTRP9, CTRP12, and CTRP13 promote an anti-inflammatory response. CTRP1 and CTRP5 are recognized to trigger inflammation and atherogenesis, whereas CTRP1 plays a protective function in insulin sensitivity and ischemic heart disease. However, the role of CTRP5 in diabetes and cardiac disease is still unclear.

**Figure 4 ijms-22-13529-f004:**
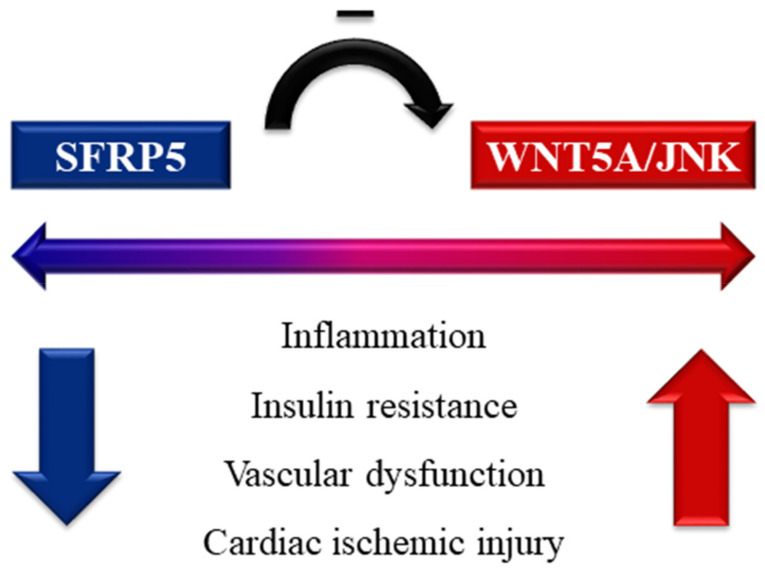
Schematic view on the association of secreted frizzled-related protein 5 (SFRP5) and wingless-type family member 5a (WNT5A)/c-Jun N-terminal kinases (JNK). WNT5A and the downstream JNK signaling facilitate inflammation and cardiometabolic impairment, which is antagonized by SFRP5.

**Table 1 ijms-22-13529-t001:** Summary of the human studies for the association of adipokine levels with cardiometabolic parameters.

	Author, Year [ref] *	Study Design ^†^	Participants	Mean Age (years)	Men (%)	Ethnicity ^‡^	Outcomes or Parameters	Cardiometabolic Health Association ^§^
Adiponectin	Gariballa et al., 2019 [35]	PC	193 overweight and obese subjects	36.0	7.0	Asian	Visceral fat	Positive
	Lindberg et al., 2014 [37]	PC	666 patients with STEMI, without diabetes	63.5	74.3	White	Incident T2DM	Positive
	Kou et al., 2018 [38]	CC	309 subjects	N/R	N/R	Asian	Atherogenic index of plasma	Positive
	Yoshida et al., 2005 [40]	CS	56 patients with T2DM	N/R	N/R	Asian	VLDL levels	Positive
	Tomono et al., 2018 [41]	CS	174 subjects without diabetes	67.9	45.4	Asian	HDL levels	Positive
	Doumatey et al., 2012 [44]	CS	822 subjects	43.3	44.3	Black	Obesity and MetS	Positive
	Garvey et al., 2014 [48]	RC	475 subjects with prediabetes and/or MetS	52.0	35.2	Multiracial	Phentermine/topiramate	Positive
	Wannamethee et al., 2007 [51]	PC	4046 men	68.7	100	White	All-cause and CVD mortality	Negative
	McEntegart et al., 2007 [52]	CS	47 patients with CAD	67.3	83.0	White	HF and cachexia	Negative
Vaspin	Taheri et al., 2020 [75]	CS	70 women	29.0	0	Asian	Obesity	Negative
	Hao et al., 2016 [77]	CS	348 subjects	52.8	58.0	Asian	T2DM and CAD	Negative
	Yang et al., 2021 [78]	CS	372 patients with T2DM	53.0	55.6	Asian	Diabetic retinopathy	Negative
	Jia et al., 2018 [79]	RC	474 patients with NAFLD	N/R	N/R	Asian	Exercise	Negative
	Tan et al., 2008 [80]	NRI	21 women with PCOS	28.0	0	White	Metformin	Negative
	Zhang et al., 2011 [81]	NRI	31 patients with T2DM	55.3	35.5	Asian	Rosiglitazone	Negative
	Golpaie et al., 2011 [82]	NRI	30 obese subjects	32.5	30.0	Asian	Restrictive bariatric surgery	Negative
	Kadoglou et al., 2021 [84]	NRI	84 subjects with preclinical carotid atherosclerosis	62.0	46.4	White	Atorvastatin	Positive
	Al-Kuraishy et al., 2018 [85]	RC	110 patients with ACS	48.6	65.5	Asian	Rosuvastatin	Positive
	Kastl et al., 2020 [86]	PC	85 patients with CAD	64.0	77.6	White	In-stent restenosis	Positive
	Zhang et al., 2016 [87]	PC	80 patients with MI	68.0	81.2	Asian	MACE	Positive
	Ji et al., 2020 [88]	RC	197 subjects with chest pain	65.0	56.9	Asian	MACE	Positive
	Rashad et al., 2020 [89]	PCC	90 patients with T2DM	58.7	55.6	Asian	IS	Negative
	Cakal et al., 2011 [90]	CS	71 women	N/R	0	Asian	Diabetogenic risk	Negative
	Esteghamati et al., 2014 [91]	CS	145 subjects	49.4	42.8	Asian	MetS	Negative
	Buyukinan et al., 2018 [92]	CS	121 obese children	N/R	34.7	Asian	MetS	Negative
CTRP9	Wang et al., 2015 [119]	CS	362 subjects	62.1	72.1	Asian	CAD	Positive
	Moradi et al., 2018 [120]	CS	337 subjects	58.0	70.0	Asian	CAD and T2DM	Negative
	Gao et al., 2019 [121]	PC	344 subjects	56.2	69.2	Asian	HFrEF	Positive
	Miyatake et al., 2020 [122]	RC	50 recipients of kidney transplantation	31.5	66.0	Asian	Aortic calcification	Positive
	Pan et al., 2020 [123]	CS	128 CI patients with cerebrovascular stent	54.0	58.6	Asian	Restenosis after cerebrovascular stent	Positive
	Jia et al., 2017 [124]	CS	306 subjects	52.0	48.0	Asian	Incident T2DM and obesity	Negative
	Hwang et al., 2014 [125]	CS	221 subjects	46.0	63.3	Asian	MetS	Positive
	Wolf et al., 2016 [126]	NRI	21 obese subjects	N/R	14.0	Multiracial	Bariatric surgery	Negative
	Xia et al., 2020 [127]	CS	259 pregnant women	N/R	0	Asian	GDM	Positive
	Yang et al., 2021 [128]	CS	262 patients with T2DM	55.0	68.3	Asian	Cardiac autonomic neuropathy	Positive
	Asada et al., 2016 [129]	CS	258 patients with T2DM without CKD	62.0	54.3	Asian	Carotid IMT	Negative
	Jung et al., 2014 [130]	CS	278 patients with T2DM	58.3	60.8	Asian	baPWV	Negative
	Na et al., 2020 [131]	CS	133 pregnant women	N/R	0	Asian	GDM	Positive
CTRP3	Zhou et al., 2018 [145]	PC	313 subjects	N/R	N/R	Asian	Incident NAFLD	Positive
	Choi et al., 2012 [146]	CS	345 subjects	51.8	38.2	Asian	Prediabetes, T2DM, and MetS	Negative
	Moradi et al., 2019 [147]	CS	164 subjects	58.0	62.8	Asian	T2DM and diabetic nephropathy	Positive
	Flehmig et al., 2014 [148]	CS	141 obese subjects	48.0	47.5	White	Metformin	Positive
	Choi et al., 2014 [151]	CS	362 subjects	60.4	67.4	Asian	ACS and SAP	Positive
	Jiang et al., 2018 [152]	CS	108 subjects	56.3	71.3	Asian	Acute aortic dissection	Positive
	Yildirim et al., 2021 [153]	CS	118 subjects	64.4	66.1	Asian	HFrEF and VT	Positive
CTRP12	Tan et al., 2014 [149]	NRI	21 women with PCOS	28.0	0	White	Metformin	Positive
	Nadimi et al., 2021 [154]	CS	250 subjects	58.5	54.8	Asian	CAD severity	Positive
CTRP13	Shanaki et al., 2016 [150]	CS	86 men	54.0	100	Asian	BMI, visceral fat, and IMT	Positive
SFRP5	Akoumianakis et al., 2019 [177]	CC	140 subjects	64.0	45.0	White	CAD	Positive
	Carstensen-Kirberg et al., 2017 [178]	CS	1096 subjects	70.2	51.5	White	Prediabetes, T2DM, BMI, and HDL-C	Positive
	Bai et al., 2021 [179]	CS	684 adolescents	13.7	54.2	Asian	FPG and TC	Positive
	Tan et al., 2014 [180]	NRI	31 obese children	11.0	71.0	Asian	Lifestyle intervention	Positive
	He et al., 2020 [181]	NRI	111 patients with T2DM	57.0	46.8	Asian	Metformin	Positive
	Hu et al., 2013 [182]	NRI	30 patients with T2DM	N/R	N/R	Asian	Liraglutide	Positive
	Almario et al., 2015 [183]	CS	84 women	36.1	0	White	Weight and cholesterol	Positive
	Xu et al., 2017 [184]	CS	284 subjects	53.4	53.9	Asian	MetS	Positive
	Lu et al., 2013 [185]	CS	124 subjects	59.5	56.5	Asian	T2DM	Negative
	Wang et al., 2021 [186]	CS	114 subjects	67.3	51.8	Asian	PAD	Positive
	Oh et al., 2020 [187]	CS	120 subjects	51.7	19.3	Asian	Vascular calcification	Positive
	Cho et al., 2018 [173]	CS	282 patients with T2DM	58.0	63.6	Asian	baPWV	Negative
	Tong et al., 2020 [188]	CS	87 subjects	61.5	56.3	Asian	CAD	Positive
	Du et al., 2019 [189]	PC	85 patients with STEMI	55.7	76.5	Asian	Early improvement of LVEF	Positive
	Wu et al., 2020 [190]	PC	833 patients with HF	65.9	57.4	Asian	Composite of all-cause mortality or HF rehospitalization	Positive
	Ji et al., 2017 [191]	PC	168 subjects	65.0	55.4	Asian	MACE	Negative
Omentin-1	Batista et al., 2007 [193]	CS	91 subjects	43.7	42.9	White	Obesity	Positive
	Zhang et al., 2014 [206]	CS	120 subjects	66.3	51.7	Asian	T2DM and obesity	Positive
	Jialal et al., 2013 [207]	CS	75 subjects	51.6	78.7	White	MetS	Positive
	Peña-Cano et al., 2021 [208]	CS	231 pregnant women	29.5	0	Hispanic	GDM	Positive
	Latif et al., 2021 [209]	CS	500 patients with T2DM	53.0	52.0	Asian	Diabetic complications	Positive
	Biscetti et al., 2019 [210]	S	600 patients with T2DM	74.7	68.2	White	PAD	Positive
	Senthilkumar et al., 2018 [211]	CS	82 patients with T2DM	48.5	N/R	Asian	Diabetic nephropathy	Positive
	El-Mesallamy et al., 2011 [212]	CS	90 subjects	57.3	74.4	Asian	T2DM	Positive
	Bai et al., 2021 [213]	CS	600 subjects	52.5	61.2	Asian	CAD	Positive
	Zhong et al., 2011 [214]	CS	207 subjects	61.2	69.1	Asian	ACS and SAP	Positive
	Biscetti et al., 2020 [215]	PC	207 patients with T2DM and CTLI	75.0	69.6	White	MACE and MALE	Positive
	Kataoka et al., 2014 [203]	CS	20 patients with acute MI	62.5	65.0	Asian	Myocardial salvage index and EF	Positive
	Narumi et al., 2014 [216]	PC	136 patients with HF	72.0	55.9	Asian	Cardiac death or HF rehospitalization	Positive
	Çelik et al., 2021 [217]	CS	121 subjects	49.6	48.8	Asian	Hypertension	Positive
	Xu et al., 2018 [218]	PC	266 patients with IS	N/R	N/R	Asian	Functional outcome	Positive
	Yang et al., 2020 [219]	PC	109 patients with CI	62.8	62.4	Asian	Functional prognosis	Positive
	Xu et al., 2018 [220]	CS	173 patients with IS	N/R	N/R	Asian	Unstable carotid plaque	Positive
	Xu et al., 2020 [221]	PC	303 patients with IS	66.8	64.7	Asian	1 year mortality	Positive
	Zhang et al., 2020 [222]	PC	104 patients with hemorrhagic stroke	68.0	54.8	Asian	Functional outcome	Positive
	Onur et al., 2014 [223]	CS	193 women	67.3	0	Asian	Angiographic CAD	Positive
	Zhou et al., 2017 [224]	CS	142 patients with CAD	N/R	N/R	Asian	Coronary collateral circulation	Positive
	Yoo et al., 2011 [225]	CS	90 subjects	54.5	41.1	Asian	T2DM and carotid plaque	Positive
	Onur et al., 2014 [226]	CS	173 subjects	N/R	N/R	Asian	PAD	Positive
	Yasir et al., 2019 [227]	CS	167 patients with T2DM	N/R	N/R	Asian	Diabetic retinopathy	Positive
	Siegrist et al., 2021 [228]	NRI	156 overweight and obese children	14.0	44.9	White	Lifestyle intervention	Positive
	Atashak et al., 2021 [229]	RC	30 obese men	N/R	100	Multiracial	Exercise	Positive
	Luis et al., 2018 [230]	NRI	24 obese subjects	N/R	N/R	White	Biliopancreatic diversion surgery	Positive
	Kadoglou et al., 2021 [84]	NRI	84 subjects with preclinical carotid atherosclerosis	62.0	46.4	White	Atorvastatin	Positive
	Alkuraishy et al., 2015 [231]	CS	85 patients with T2DM and acute MI	57.5	60.0	White	Metformin	Positive

* Articles were listed in the order of citation in the text. Meta-analyses were excluded. ^†^ RC, randomized controlled; NRI, nonrandomized interventional; PC, prospective cohort; RC, retrospective cohort; CC, case-control; PCC, prospective case-control; CS, cross-sectional. ^‡^ One of the following ethnic groups: white, Asian, Hispanic, black, and multiracial. ^§^ Labeled as positive if the increasing adipokine levels are associated with the positive direction of cardiometabolic health, and as negative in the case of the inverse association.

## Data Availability

Not applicable.

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
