# Peer review of "The Role of Anti-Inflammatory Adipokines in Cardiometabolic Disorders: Moving beyond Adiponectin"

_ijms, 2021, doi:10.3390/ijms222413529_

Round 1

Reviewer 1 Report

The review is informative and well written. It adequately summarizes current evidence on the topic and directs the way for new studies.

However, there are a major issue to raise. Omentin-1 is not mentioned in the manuscript. There are many evidences in the literature of its anti-inflammatory effects and its link with cardiovascular disease. In particular, there is at least one prospective clinical study demonstrating that reduced levels of omentin-1 are associated with major adverse cardiovascular events (MACE) in patients with peripheral artery disease. I suggest to cite recent and important papers, available in literature, focused on biomarkers of vascular disease in diabetes, in particular about the adipose-immune system cross talk.

Reviewer 2 Report

In this review article titled " The role of anti-inflammatory adipokines in cardiometabolic disorders: moving beyond Adiponectin" have emphasized the role of anti-inflammatory function of adipose tissue-derived cytokines such as adipokines beyond it pro-inflammatory action in cardiometabolic disorders including obesity, T2DM, liver disease and atherosclerosis which is new and fascinating area of research. Considering the increasing incidence of cardiometabolic diseases and its association with obesity and lack of direct therapeutics has opened new front for the development of new drugs for cardiometabolic diseases. Thus, the consideration of obesity based biomarkers for early detection of cardiometabolic diseases have great potential for the development of new drug against metabolic disease.  In this article authors have discussed adipose derived adipokines as a  anti-inflamamtory markers in the protection of cardiometabolic disease. Although this review has discussed only association studies, however, detailed mechanistic studies should be discussed.    

Minor revison with typo error. 

Reviewer 3 Report

The authors provide a review on the evidence on the role of anti-inflammatory adipokines, namely, adiponectin, vaspin, the C1q/TNF-related protein (CTRP) family, and secreted frizzled-related protein 5 (SFRP5) in obesity/cardiometabolic disorders. It was concluded that further investigation on the mechanism of action and prospective human trials may pave the way to their clinical application as innovative biomarkers and therapeutic targets. This is a very well written review, it is well structured and the figures are clear and informative. Pertinent references have been cited. However, I have a few recommendations for consideration.

  1. Abstract.  Lines 14-16. It would be good if the authors can add and complete the sentence as to what condition the role of adipokines is being described In addition, for the sentence beginning line 16 and ending on 18, again as a concluding statement it would be good if the authors can complete the sentence and state which disease condition the clinical applicability is indicated.
  2. A summary table of the human studies would be of benefit to the reader and perhaps help in understanding the disparities in human studies, For each study, patient/participant type, age, sex, ethnicity and outcomes could be indicated in this table.
  3. Similarly, if there is information available in the literature, and given the role of ethnicity, age and sex in obesity, can the authors discuss these elements in relation to adipokine biomarkers and clinical applicability?
  4. The authors should elaborate and emphasize on the novelty aspect of their review in relation to other similar published reviews. 

Round 2

Reviewer 1 Report

The authors significantly modified the manuscript that is now suitable for publication.

This manuscript is a resubmission of an earlier submission. The following is a list of the peer review reports and author responses from that submission.

Round 1

Reviewer 1 Report

major comments

  1. Classical role of adiponectin is not necessary in this article as fitting the tile of beyond adiponectin. My opinion is to shorten the classical role of adiponectin part as possible and extend more about the latter part with anti-diabetic agents and adiponectin or more recent advances in adiponectin part such a relationship with FGF 21  etc. We need more recent updates.
  2. Vaspin as a mechanism is understandable as an anti-inflammatory adipokines, however, please show more data in humans. The current values and levels of vaspin in human metabolic diseases or the part we need for the future research or evidences, and clinical application. 
  3. CTRP part is also too long for the mechanism part. Please make more impact arrangement for explanations of basic studies and find more explanation of human studies and its value as a biomarker or predictor for certain metabolic diseases. There are also several data of C1q-adiponectin complex in human metabolic diseases.
  4. Please expand the conclusion, or make more proactive suggestions of those adipokines through this review: have what kinds of clinical meaning or as a biomarkers for the readers 

minor comments

  1. For example, ref 17, ref 18 was marked as twice in the quoted sentences from same article. Please make more arrangement of simple reference marking in the paragraphs.

Reviewer 2 Report

The manuscript is too superficial, generic and summary. The topic is not deepened, the figures generated only for some adipokines.

The biological effects (metabolic, inflammatory, cardiovascular) of the adipokines are reported in the same paragraph without focus on what is the title and topic of the manuscript.

Evidence in vitro, in vivo and in animal models is reported in a confusing manner.

Round 2

Reviewer 1 Report

Authors made this revised manuscript with reflecting my opinions. I do not have more comments and hope this article can give informative idea to the readers.

Author Response

The reviewer has accepted the article. The authors appreciate the reviewer's comments. 

Reviewer 2 Report

the authors have improved the manuscript but stil I do believe that is not acceptable for pubblication.

  • title: why bejond adiponectin? the authors give the same attention to alla adipokines
  • why figures only for 2 dipokines?
  • biological functions of adipokines respect to what? cardiometabolic disorders? the reported adipokines have functions wider than what the authors present
  • in human relevance there are stil reported facts referred to animal studies while most of aniaml studies have been removed (why?)
  • in general, I do not believe that this manuscript adds a significant contribution to the state of the art. where is the novelty of this manuscript compared to already published reviews?  

Round 3

Reviewer 2 Report

- the manuscript is not enough novel and informative to be acceptable. For example, some other relevant adipokines (i.e. leptin and resistin) have not been analyzed. secondly, cardiometabolic diseases is very general, the authors do not give information about different diseases. thirdly, the molecular pathways induced by the adipokines have not been deeply investigated.